# Joint-Motion Mutual Learning for Pose Estimation in Videos

Sifan Wu[*]
College of Computer Science and Technology, Jilin University, Changchun, China
wusifan2021@gmail.com

Haipeng Chen[*][†]
College of Computer Science and Technology, Jilin University, Changchun, China
chenhp@jlu.edu.cn

Yifang Yin
Institute for Infocomm Research ($I^2R$), A*STAR, Singapore
yin_yifang@i2r.a-star.edu.sg

Sihao Hu[†]
Georgia Institute of Technology, Atlanta, USA
sihaohu@gatech.edu

Runyang Feng
School of Artificial Intelligence, Jilin University, Changchun, China
runyang2019.feng@gmail.com

Yingying Jiao[*][†]
College of Computer Science and Technology, Jilin University, Changchun, China
jiaoyy21@mails.jlu.edu.cn

Ziqi Yang[‡]
The State Key Laboratory of Blockchain and Data Security, Zhejiang University, Hangzhou, China
yangziqi@zju.edu.cn

Zhenguang Liu[†][‡]
The State Key Laboratory of Blockchain and Data Security, Zhejiang University, Hangzhou, China
liuzhenguang2008@gmail.com

## Abstract

Human pose estimation in videos has long been a compelling yet challenging task within the realm of computer vision. Nevertheless, this task remains difficult because of the complex video scenes, such as video defocus and self-occlusion. Recent methods strive to integrate multi-frame visual features generated by a backbone network for pose estimation. However, they often ignore the useful joint information encoded in the initial heatmap, which is a by-product of the backbone generation. Comparatively, methods that attempt to refine the initial heatmap fail to consider any spatio-temporal motion features. As a result, the performance of existing methods for pose estimation falls short due to the lack of ability to leverage both local joint (heatmap) information and global motion (feature) dynamics.

To address this problem, we propose a novel joint-motion mutual learning framework for pose estimation, which effectively concentrates on both local joint dependency and global pixel-level motion dynamics. Specifically, we introduce a context-aware joint learner that adaptively leverages initial heatmaps and motion flow to retrieve robust local joint feature. Given that local joint feature and global motion flow are complementary, we further propose a progressive joint-motion mutual learning that synergistically exchanges information and interactively learns between joint feature and motion flow to improve the capability of the model. More importantly, to capture more diverse joint and motion cues, we theoretically analyze and propose an information orthogonality objective to avoid learning redundant information from multi-cues. Empirical experiments show our method outperforms prior arts on three challenging benchmarks.

## CCS Concepts

• **Computing methodologies → Interest point and salient region detections**.

## Keywords

Pose estimation, Human pose estimation, Multi-person, Multi-frame, Mutual learning

**ACM Reference Format:**
Sifan Wu, Haipeng Chen, Yifang Yin, Sihao Hu, Runyang Feng, Yingying Jiao, Ziqi Yang, and Zhenguang Liu. 2024. Joint-Motion Mutual Learning for Pose Estimation in Videos. In *Proceedings of the 32nd ACM International Conference on Multimedia (MM '24), October 28-November 1, 2024, Melbourne, VIC, Australia.* ACM, New York, NY, USA, 10 pages. https://doi.org/10.1145/3664647.3681179

[*]Also with Key Laboratory of Symbolic Computation and Knowledge Engineering of Ministry of Education, Jilin University, Changchun, China.

[†]Corresponding Authors: Yingying Jiao, Haipeng Chen, Zhenguang Liu, Sihao Hu.

[‡]Also with Hangzhou High-Tech Zone (Binjiang) Institute of Blockchain and Data Security, Hangzhou, China.

## 1 Introduction

Accurately estimating human pose in videos by localizing the anatomical position of human joints (e.g., ankle, shoulder) serves as a fundamental cornerstone for higher-level tasks such as action recognition [62], motion transfer [36, 58], and motion prediction [7, 47]. It also paves the way for modern human-machine interaction applications, including robotics [32], augmented reality (AR) [19], and smart home [46], *etc.*

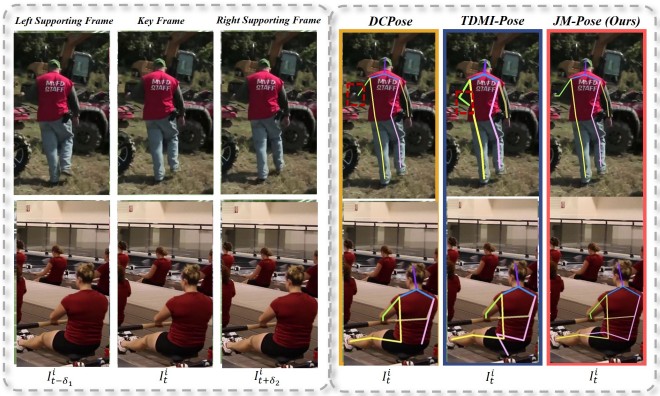

**Figure 1: Heatmap-based methods such as DCPose refine heatmaps to perform pose estimation but lack the ability to incorporate spatio-temporal features. Conversely, feature-based methods like TDMI employ feature difference to extract the valuable features but neglect local joint dependency present in heatmaps. These methods may encounter estimation difficulties in scenarios involving video defocus and self-occlusion. In comparison, our approach models the joint-motion information with a novel mutual learning framework, which grasps more meaningful and complementary representations, delivering a more robust result.**

Fueled by the power of deep learning [9, 44, 65, 66, 71] and the accessibility of large-scale datasets [1, 10, 23], human pose estimation in static images has garnered extensive attention and achieved notable breakthroughs. Pioneering approaches propose pictorial structure models [43, 54] for pose estimation. Recently, deep convolutional neural models [33, 52], Transformer models [61, 67], and diffusion models [40] have drastically pushed forward the frontier of pose estimation in static images. Unfortunately, existing methods often encounter suboptimal performance when directly applied to human pose estimation in videos. This mainly stems from the fact that videos are usually accompanied with challenging frames due to frequent pose occlusion and video blurring.

Existing methods for video-based human pose estimation can be broadly classified into heatmap-based and feature-based approaches. These methods are distinguished by their emphasis on refining either *heatmaps* or *visual features* extracted from the backbone network. One line of works, *i.e., heatmap-based* methods [5, 34], utilize unidirectional or bidirectional neighboring heatmaps of consecutive frames to refine the heatmap of the keyframe. Another line of works, *i.e., feature-based* approaches [14, 35], focus on extracting the most useful visual features for human pose estimation.

Upon scrutinizing and experimenting with the released implementations of existing methods [14, 34], we observe that they suffer from degraded performance in challenging scenes like video defocus and self-occlusion: (1) For example, in Figure 1, the heatmap-based method DCPose [34] fails to locate the left hand in a defocused frame and the right foot in a self-occlusion frame. The reason is that DCPose estimates the joint by merging the initial heatmaps extracted from consecutive frames and ignores the cues contained

in the spatio-temporal motion feature. (2) As shown in Figure 1, the feature-based method TDMI [14] demonstrates inaccuracies in estimating joint positions within challenging frames. These include the misalignment of the left hand in a defocused frame and the drifting right foot in a self-occlusion frame. We think that TDMI employs joint (heatmap)-agnostic visual features to estimate human pose, overlooking the implicit position information and joint dependency contained in initial heatmaps. In addition, visual features fail to capture pixel-level motion dynamics and are not robust enough for challenging scenarios. These observations inspire us to ask: *How to effectively aggregate **local joint dependency** and **global pixel-level motion dynamics** to obtain a comprehensive representation?*

To answer the question, we propose a novel framework, termed **J**oint-**M**otion Mutual Learning for **Pose** Estimation (**JM-Pose**), which comprises two key components. *Firstly*, to enable the model to learn joint (heatmap)-relevant features, we introduce *a context-aware joint learner* to extract local joint feature. Specifically, guided by initial heatmaps, we employ modulated deformable operations to capture local joint context features. *Secondly*, we propose *a progressive joint motion mutual learning* to explore the cooperative effect of the two representations in pose estimation. This alleviates the limitation of existing methods that only exploit single-modal representations. The progressive joint motion mutual learning dynamically exchanges and refines information between local joint feature and global motion flow to capture an informative joint-motion representation. We conduct further theoretical analysis and arrive at an information orthogonality objective. Minimizing the objective endows our model with the ability to fully mine the diverse joint feature and motion flow, enhancing the performance of pose estimation. We perform comprehensive experiments on three widely used benchmark datasets. Empirical results demonstrate that our method consistently outperforms state-of-the-art approaches.

**Contribution.** In summary, this paper makes three original contributions:

- To the best of our knowledge, we are the first to present a joint-motion mutual learning framework for human pose estimation, which collaboratively amalgamates local joint feature and global pixel-level motion flow.
- We propose a novel context-aware joint learner guided by heatmaps to retrieve refined local joint-level information from the motion flow. We also learn a comprehensive joint-motion representation by progressive joint-motion mutual learning with information orthogonality objective, which effectively captures the diverse knowledge from complementary features.
- Extensive experiments on three widely used benchmark datasets show that our method achieves a new state-of-the-art performance.

## 2 Related Work

### 2.1 Image-based Human Pose Estimation

Before the surge of deep learning, traditional approaches [49, 56, 63] address the human pose estimation task by designing a probabilistic graphical structure or a pictorial structure to model the relationship of human joints. With the advent of deep learning such as CNN

[29], Transformer [53], diffusion model [21] and the large-scale datasets such as PoseTrack [1, 10, 23], human pose estimation in static images has been extensively studied and achieved significant breakthroughs [2, 31, 57]. There are two mainstream paradigms: bottom-up and top-down. Bottom-up methods [27, 28, 38] are also known as part-based methods. They first detect all human joints in the given images and then assemble them into individual skeletons. [6] employs a dual branch structure and utilizes part affinity fields to represent pairwise confidence maps between different body parts. [37] assigns an identification tag to each detected part that indicates to which individual it belongs. Although bottom-up methods have obtained promising results, their human part detectors are fragile when the people in the image are far away from the camera position. Contrary to the above method, top-down methods [12, 13, 48] first detect all human bounding boxes in the given images and then perform pose estimation on each individual. [48] proposes a novel framework, named HRNet, which preserves high-resolution features through the multiple stages for precise human pose estimation. [8] proposes a feature pyramid network to detect simple joints and a fine-grained network to identify hard joints which incorporate all the hierarchical representations of the previous network.

## 2.2 Video-based Human Pose Estimation

Applying image-level approaches to video-level pose estimation results in degraded performance due to the neglect of temporal cues inherent in videos. Current methods for video-level human pose estimation can be classified into heatmap-based methods and feature-based methods, depending on whether the approach focuses on refining either *heatmaps* or *visual features* extracted from the backbone network.

The heatmap-based methods could employ optical flow fields, temporal models (*i.e.*, RNN and Transformer), or heatmap optimization to detect human pose. To capture temporal dynamics in the video, (i) [45, 69] compute dense optical flow between consecutive frames for human pose estimation. However, the computation of optical flow is expensive and largely depends on the quality of the video, and challenging cases such as blurring and character occlusion are very common in videos. (ii) [2] employs an LSTM module to utilize the similarities and temporal relationships between frames, achieving promising results. However, RNNs have limitations in focusing on global temporal information, leading to the development of Transformer-based approaches. [16] propose using a Transformer to model spatiotemporal representation with self-feature refinement and cross-frame temporal learning. (iii) [5, 34] employ unidirectional or bidirectional neighboring heatmaps of consecutive frames to refine the heatmap of the keyframe.

The feature-based method tends to fuse meaningful features to perform pose estimation. (i) [35] proposes feature alignments between supporting frames and the keyframe and utilizes the aligned feature to estimate human pose. (ii) [14] employs feature differences and disentanglement to obtain useful visual features to detect human poses. However, visual features contains much task-irrelevant information, such as background and nearby-person. We employ pixel-level optical flow to capture global motion dynamic, which is more robust to challenging scenes.

## 3 Method

**Preliminaries.** Presented with a video, our task is to estimate human poses in each frame. Technically, all human bounding boxes in each frame are identified with the object detector proposed in [39]. These bounding boxes are then enlarged by 25% to crop the identical human across consecutive frames. Subsequently, for person $k$, we acquire the cropped video clip $\mathcal{I}_t^k = < I_{t-\delta}^k, \ldots, I_t^k, \ldots, I_{t+\delta}^k >$ centered on the key frame $I_t^k$ (where $\delta$ represents the predefined time span), and aim to estimate the pose of person $k$ in frame $I_t^k$. For simplicity, we set $\delta = 1$ and omit the superscript $k$.

**Method overview.** An overview of our approach, JM-Pose, is outlined in Figure 2. JM-Pose consists of two key components: a Context-aware Joint Learner (CJL, Subsec. 3.1) and a progressive Joint-Motion Mutual learning module (JMML, Subsec. 3.2). Formally, we first extract initial heatmaps $\hat{H}_t$ and pixel-level motion flow $M_t$ from the input sequence $\mathcal{I}_t$, and feed them into the context-aware joint learner. The CJL utilizes modulated deformable operations to retrieve the local joint feature $J_t$ from motion flow. Subsequently, the progressive joint-motion mutual learning module accepts local joint feature $J_t$ and global motion flow $M_t$ as inputs, and dynamically exchanges information between them. This yields a more informative joint-motion representation $S_t$. Finally, $S_t$ is fed into a detection head to generate the human pose estimation $\hat{\mathcal{H}}_t$. In what follows, we will elaborate on the two key components in detail.

## 3.1 Context-Aware Joint Learner

In order to better leverage local joint-related contexts, we propose to explicitly retrieve the local joint feature from the consecutive frames with the guidance of readily available initial heatmaps. There are two key steps: feature extraction and deformable feature retrieval.

**Feature extraction.** Specifically, given the video clip $\mathcal{I}_t$, we first employ a backbone [48] network to generate initial heatmaps $\mathbb{H} = < H_{t-1}, H_t, H_{t+1} >$, which encode possible position information of the joints in each frame. Then, these heatmaps are aggregated to yield the sequence-level representations for the current frame $\hat{H}_t$. In parallel to the heatmap extraction, we also employ RAFT [50] to extract dense motion flow $M_t$, which captures the spatial variation of joint pixel positions over time, *i.e.*, the dynamic information of the pose, as follows:

$$M_t = \Theta(I_{t-1}, I_t) \oplus \Theta(I_t, I_{t+1}), \tag{1}$$

where $\oplus$ represents the concatenation operation and $\Theta(\cdot, \cdot)$ represents the RAFT estimator.

**Deformable feature retrieval.** Given $\hat{H}_t$ and $M_t$, we further retrieve context-aware joint cues from motion flow $M_t$ guided by the initial heatmap $\hat{H}_t$.

Instead of fusing $M_t$ and $\hat{H}_t$ with regular convolution blocks, we employ modulated deformable convolution to extract the context-aware joint feature $J_t$, sampling from motion flow by adaptively learning the offsets and weights of the deformable convolution. Initially, we concatenate the motion flow $M_t$ and the heatmap $\hat{H}_t$ to generate the fused features $R_t$. Subsequently, given the fused features $R_t$, we compute the kernel sampling offsets $JO_t$ and modulated weights $JW_t$ by:

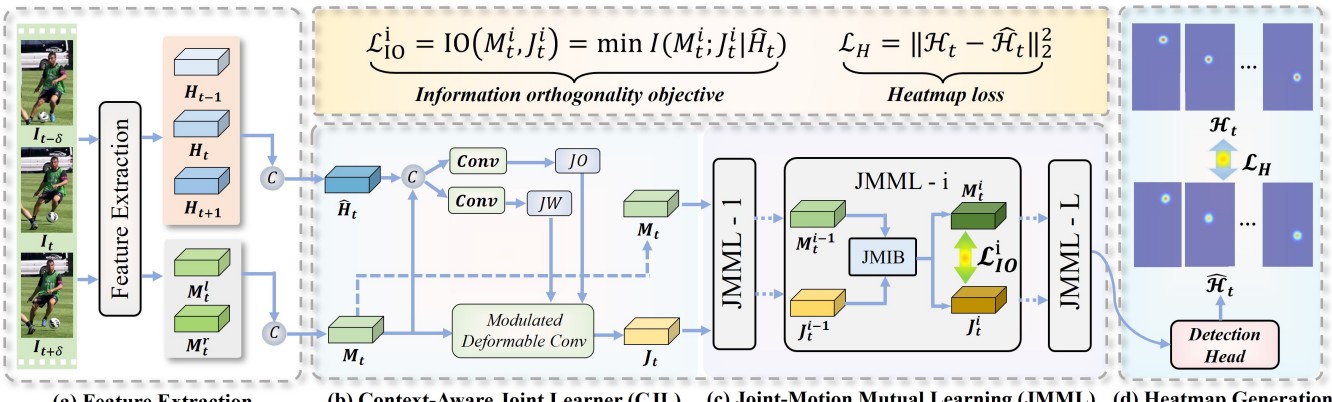

**(a) Feature Extraction**     **(b) Context-Aware Joint Learner (CJL)**     **(c) Joint-Motion Mutual Learning (JMML)**     **(d) Heatmap Generation**

**Figure 2: JM-Pose is designed to estimate human pose in the keyframe $I_t$ with its consecutive supporting frames, *e.g.*, $I_{t-\delta}, I_{t+\delta}$ in the above figure. JM-Pose introduces two key components: *Context-aware Joint Learner* and *Joint-Motion Mutual Learning*. The context-aware joint learner is designed to extract the local joint-level feature $J_t$ from motion flow $M_t$ using modulated deformable operations guided by initial heatmap $\hat{H}_t$. Joint-motion mutual learning further refines local joint feature $J_t$ and global motion flow $M_t$ using their knowledge to complement each other. An information orthogonality objective $\mathcal{L}_{IO}$ is adopted to improve the diversity of learned $J_t$ and $M_t$, which is conditioned on initial heatmap $\hat{H}_t$. The final $L^{th}$ representation is aggregated and fed to the detection head to obtain the final heatmap $\hat{\mathcal{H}}_t$ for pose estimation. Finally, we employ a heatmap loss $\mathcal{L}_H$ to measure the discrepancy between the ground truth $\mathcal{H}_t$ and the detected heatmaps $\hat{\mathcal{H}}_t$.**

$$\{\hat{H}_t \oplus M_t\} \xrightarrow[blocks]{convolution} R_t, \qquad (2)$$

$$R_t \xrightarrow[blocks]{residual} \xrightarrow[blocks]{convolution} JO_t, \qquad (3)$$

$$R_t \xrightarrow[blocks]{residual} \xrightarrow[blocks]{convolution} JW_t, \qquad (4)$$

where the kernel offset $JO_t$ reflects the pixel movement association field, and the modulated weight $JW_t$ represents the scalar matrix for the convolution kernel. The network parameters of the two processes for $JO_t$ and $JW_t$ are learned independently. Finally, we retrieve the context-aware joint feature $J_t$ via a modulated deformable convolution [72] as follows:

$$\{M_t, JO_t, JW_t\} \xrightarrow[convolution]{deformable} J_t. \qquad (5)$$

## 3.2 Joint-Motion Mutual Learning

Context-aware joint learner outputs the joint feature $J_t$ that encodes local joint dependency and the motion flow $M_t$ that encodes global pixel-level motion dynamics. Simply merging these two features may not fully exploit their individual strengths for accurate pose estimation, since joint feature and motion flow essentially have the potential to guide each other. For instance, in scenes with video defocus (poor video quality), local joint feature can supply motion flow with position information for hard-to-detect joints. Conversely, in scenarios of pose occlusion, the occluded joints could be inferred based on global motion dynamics. Given that motion flow and joint feature are complementary, we propose a progressive L-layer joint-motion mutual learning (JMML) paradigm. This process involves a joint-motion interaction block and an information

orthogonality objective, aimed at effectively enhancing the expressive capabilities of each feature by densely exchanging valuable information between them.

**Joint-motion interaction block (JMIB).** Here, we elaborate on the $i^{th}$ layer joint-motion interaction block in JMML. As shown in Figure 3, with $J_t^{i-1}$ and $M_t^{i-1}$ produced by $i-1^{th}$ JMML, $i^{th}$ joint-motion interaction block applies two convolution blocks on them, obtaining $\hat{J}_t^{i-1}$ and $\hat{M}_t^{i-1}$. Subsequently, to leverage the complementarity of local joint feature and global motion flow, we propose a joint-motion cross-attention mechanism within the interaction block, which allows joints and motion features to act as mutual guides. Specifically, we generate a joint query matrix $Q^j$ based on $\hat{J}_t^{i-1}$ while also generating a motion key matrix $K^m$ and a motion value matrix $V^m$ based on $\hat{M}_t^{i-1}$ using independent convolution layers. Similarly, we employ $\hat{M}_t^{i-1}$ to generate a motion query matrix $Q^m$ while also employing $\hat{J}_t^{i-1}$ to generate a joint key matrix $K^j$ and a joint value matrix $V^j$. Then, we compute feature compatibility between corresponding query-key pairs using a cross attention mechanism:

$$\hat{J}_t^{i-1} = \Upsilon(J_t^{i-1}), \quad \hat{M}_t^{i-1} = \Upsilon(M_t^{i-1}), \qquad (6)$$

$$\tilde{J}_t^{i-1} = \Psi(Q^j K^{mT}/\sqrt{d})V^m + \hat{J}_t^{i-1}, \qquad (7)$$

$$\hat{M}_t'^{i-1} = \Psi(Q^m K^{jT}/\sqrt{d})V^j + \hat{M}_t^{i-1}, \qquad (8)$$

where $\Upsilon(\cdot)$ represents a set of convolution blocks, $\Psi(\cdot)$ denotes the softmax operation, and $d$ is the hyperparameter. Joint-motion cross-attention coarsely instills local joint feature and global motion flow. Next, we will exchange each other's information finely. Consider the complementarity of the two features, we concatenate $\tilde{J}_t'^{i-1}$ and $\hat{M}_t'^{i-1}$ followed by a sigmoid function and a channel chunk

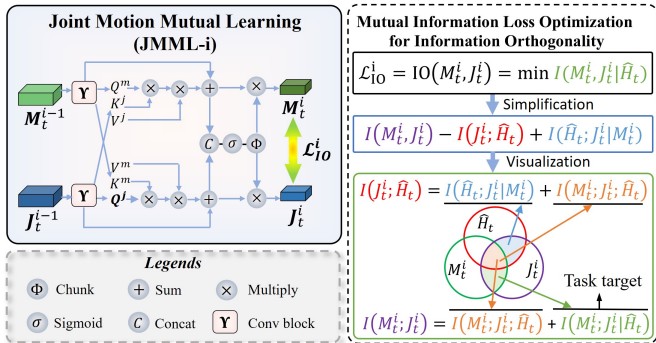

**Figure 3: The joint-motion mutual learning framework. Left: The architectures of the $i^{th}$ joint-motion mutual learning and legends. Right: We propose an information orthogonality objective to update the parameters of joint-motion mutual learning and mine diverse local joint feature and global motion flow.**

operation to obtain two attention masks $A^j$ and $A^m$. The attention mask $A^j$ and $A^m$ are used to rescale corresponding joint feature $\hat{J}_t'^{i-1}$ and motion flow $\hat{M}_t'^{i-1}$ respectively. The above process can be described as :

$$A^j, A^m = \Phi(\sigma(\hat{J}_t'^{i-1} \oplus \hat{M}_t'^{i-1})), \tag{9}$$

$$J_t^i = A^j \otimes \hat{J}_t^{i-1}, M_t^i = A^m \otimes \hat{M}_t^{i-1}, \tag{10}$$

where $\Phi(\cdot)$ denotes the operations of channel chunk, $\oplus$, $\otimes$, and $\sigma(\cdot)$ reveal concatenate, spatial-wise multiplication, and the sigmoid activation function, respectively.

**Information orthogonality objective.** After generating the complementary joint feature $J_t^i$ and motion flow $M_t^i$, we further conduct theoretical analysis and propose an information orthogonality objective inspired by mutual information theory [20]. This objective ensures that both joint feature and motion flow provide diverse cues by minimizing redundant information introduced by the mutual learning block.

Mutual information measures the information dependency between random variables, which can be formalized as:

$$\mathcal{I}(x_1; x_2) = \mathbb{E}_{p(x_1,x_2)} [\log \frac{p(x_1, x_2)}{p(x_1)p(x_2)}], \tag{11}$$

where $x_1$ and $x_2$ denote the two random variables, $p(x_1, x_2)$ represents the joint probability distribution between $x_1$ and $x_2$. $p(x_1)$ and $p(x_2)$ represent the marginals of $x_1$ and $x_2$, respectively.

To ensure the diversity of features extracted from the mutual learning ($J_t^i$ and $M_t^i$) and information gain of the initial heatmap $\hat{H}_t$, the proposed information orthogonality loss is given by:

$$\mathcal{L}_{IO}^i = IO(M_t^i, J_t^i) = min\ \mathcal{I}(M_t^i; J_t^i | \hat{H}_t). \tag{12}$$

As $\mathcal{L}_{IO}^i$ in equation 12 is intractable to estimate analytically, we seek to simplify it as depicted in Figure 3. The information orthogonality loss $\mathcal{L}_{IO}^i$ can be stated as :

$$min\ \mathcal{I}(M_t^i; J_t^i | \hat{H}_t) = min\ [\underbrace{\mathcal{I}(M_t^i; J_t^i)}_{relevancy} - \overbrace{\mathcal{I}(J_t^i; \hat{H}_t)}^{complement} + \underbrace{\mathcal{I}(\hat{H}_t; J_t^i | M_t^i)}_{redundancy}]. \tag{13}$$

The first term compresses the relevancy between joint feature $J_t^i$ and motion flow $M_t^i$, directing both to focus on more diverse cues. The second term allows the joint feature $J_t^i$ to contain more location and dependency information related to the heatmap $\hat{H}_t$. The third term constrains the context-aware joint learner to extract joint feature $J_t$ from the motion flow $M_t$. These terms are illustrated in Figure 3 and can be estimated by existing mutual information estimators [35, 51]. In our experiments, we employ the Variational Self-Distillation [51] to estimate each term.

**Heatmap estimation.** Ultimately, we derive the joint feature $J_t^L$ and motion flow $M_t^L$ through the $L^{th}$ joint-motion mutual learning. We aggregate these two features to obtain a comprehensive representation $S_t$ as follows:

$$S_t = \Upsilon(J_t^L \oplus M_t^L), \tag{14}$$

where $\Upsilon(\cdot)$ denotes a convolution block and $\oplus$ is the concatenation operation. $S_t$ is subsequently fed into a detection head to generate the pose heatmap $\hat{\mathcal{H}}_t$. The detection head consists of a set of $3 \times 3$ convolution layers. Through mutual and adaptive refinement of local joint dependency and global pixel-level motion flow, we can obtain complementary cues between them, thereby generating more tailored information that contributes to accurate pose estimation.

### 3.3 Loss Functions

We adopt a standard heatmap loss $\mathcal{L}_H$ to supervise the final pose estimation:

$$\mathcal{L}_H = \left\| \mathcal{H}_t - \hat{\mathcal{H}}_t \right\|_2^2, \tag{15}$$

where $\mathcal{H}_t$ and $\hat{\mathcal{H}}_t$ represent the ground truth heatmap and the prediction heatmap, respectively. We also employ the proposed *information orthogonality loss* to supervise the joint-motion mutual learning as mentioned in Sec. 3.2. The total loss function $\mathcal{L}_{Total}$ is given by:

$$\mathcal{L}_{Total} = \mathcal{L}_H + \alpha \sum_{i=1}^{L} \mathcal{L}_{IO}^i, \tag{16}$$

where $\alpha$ is a trade-off parameter to balance the two loss terms.

## 4 Experiments

In this section, we conduct comprehensive experiments on three widely used benchmark datasets for human pose estimation in video. Our overarching goal is to answer the following research questions:

- RQ1: How does our method compare with the state-of-the-art human pose estimation approaches on **quantitative** results?

- RQ2: How is the proposed method compared against existing human pose estimation methods in **challenging scenarios**?
- RQ3: How does the proposed method compare to the state-of-the-art video human pose estimation methods on **visual** results?
- RQ4: How much do various components of JM-Pose contribute to its overall performance?

Next, we first introduce the experimental settings, followed by answering the above research questions one by one.

### 4.1 Experimental Settings

**Datasets.** We adopt three widely-used pose estimation datasets, namely PoseTrack2017 [23], PoseTrack2018 [1], and PoseTrack21 [10], to evaluate the proposed method.

PoseTrack is a large-scale public dataset designed for evaluating human pose estimation methods in video and involves the challenging case of complex movements of highly occluded people in crowded scenes. Specifically, **PoseTrack2017** comprises 300 video sequences and 80,144 pose annotations, with 250 video clips allocated for training and the remaining used for validation (split following the official protocol). **PoseTrack2018** increases the number of videos and pose annotations, including 593 video clips for training and 170 video clips for validation (with 153,615 pose labels). Each annotated person in both datasets has 15 keypoints positions and a flag indicating joint visibility. Moreover, the training samples provide dense labels in the center 30 frames, and validation videos are additionally labeled every four frames. The **Posetrack21** further extends PoseTrack2018 with a focus on small subjects and those in the occluded scenes, encompassing 177,164 pose labels.

**Evaluation metric.** The average precision (**AP**) is commonly adopted to evaluate the performance for human pose estimation. We further average the AP of all joints (**mAP**) to derive and compare the final performance.

**Implementation details.** We use PyTorch to implement our method. The input image size is fixed to $384 \times 288$. For initial heatmap extraction, we leverage the HRNet-W48 model pre-trained on the COCO dataset as our backbone. During the training process, we incorporate several data augmentations including random rotation $[-45°, 45°]$, random scale $[0.65, 1.35]$, random truncation, and flipping. The predefined time span $\delta$ is set to 2. In joint-motion mutual learning, $L$ is set to 4. We utilize the AdamW optimizer with an initial learning rate of $1e-4$ (decays to $1e-5, 1e-6, 1e-7$ at the $5^{th}, 10^{th}, and\ 15^{th}$ epochs, respectively). Our model is trained on 2 Nvidia Geforce RTX 4090 GPUs and terminated within 20 epochs.

### 4.2 Quantitative Comparison with State-of-the-art Methods (RQ1)

**Results on the PoseTrack2017 dataset.** We first evaluate the performance of our model on the PoseTrack2017 validation dataset. Specifically, we compare 22 methods, and their performance are shown in Table 1. The proposed JM-Pose consistently outperforms existing approaches, achieving an mAP of 86.4. In comparison to the widely-used backbone network HRNet, our JM-Pose improves the mAP by 9.1 points. Moreover, when compared to the state-of-the-art method TDMI, JM-Pose delivers a 0.7 mAP gain. Notably, we observe an encouraging improvement for the challenging joints

**Table 1: Quantitative results on the PoseTrack2017 dataset.**

| Methods | Head | Shoulder | Elbow | Wrist | Hip | Knee | Ankle | Mean |
|---|---|---|---|---|---|---|---|---|
| PoseTracker [17] | 67.5 | 70.2 | 62.0 | 51.7 | 60.7 | 58.7 | 49.8 | 60.6 |
| PoseFlow [60] | 66.7 | 73.3 | 68.3 | 61.1 | 67.5 | 67.0 | 61.3 | 66.5 |
| JointFlow [11] | - | - | - | - | - | - | - | 69.3 |
| FastPose [70] | 80.0 | 80.3 | 69.5 | 59.1 | 71.4 | 67.5 | 59.4 | 70.3 |
| TML++ [22] | - | - | - | - | - | - | - | 71.5 |
| Simple (R-50) [59] | 79.1 | 80.5 | 75.5 | 66.0 | 70.8 | 70.0 | 61.7 | 72.4 |
| Simple (R-152) [59] | 81.7 | 83.4 | 80.0 | 72.4 | 75.3 | 74.8 | 67.1 | 76.7 |
| STEmbedding [26] | 83.8 | 81.6 | 77.1 | 70.0 | 77.4 | 74.5 | 70.8 | 77.0 |
| HRNet [48] | 82.1 | 83.6 | 80.4 | 73.3 | 75.5 | 75.3 | 68.5 | 77.3 |
| MDPN [18] | 85.2 | 88.5 | 83.9 | 77.5 | 79.0 | 77.0 | 71.4 | 80.7 |
| CorrTrack [42] | 86.1 | 87.0 | 83.4 | 76.4 | 77.3 | 79.2 | 73.3 | 80.8 |
| Dynamic-GNN [64] | 88.4 | 88.4 | 82.0 | 74.5 | 79.1 | 78.3 | 73.1 | 81.1 |
| PoseWarper [5] | 81.4 | 88.3 | 83.9 | 78.0 | 82.4 | 80.5 | 73.6 | 81.2 |
| DCPose [34] | 88.0 | 88.7 | 84.1 | 78.4 | 83.0 | 81.4 | 74.2 | 82.8 |
| DetTrack [55] | 89.4 | 89.7 | 85.5 | 79.5 | 82.4 | 80.8 | 76.4 | 83.8 |
| SLT-Pose [16] | 88.9 | 89.7 | 85.6 | 79.5 | 84.2 | 83.1 | 75.8 | 84.2 |
| HANet [25] | 90.0 | 90.0 | 85.0 | 78.8 | 83.1 | 82.1 | 77.1 | 84.2 |
| KPM [15] | 89.5 | 90.9 | 87.6 | 81.8 | 81.1 | 82.6 | 76.1 | 84.6 |
| M-HANet [24] | 90.3 | 90.7 | 85.3 | 79.2 | 83.4 | 82.6 | 77.8 | 84.8 |
| FAMI-Pose [35] | 89.6 | 90.1 | 86.3 | 80.0 | 84.6 | 83.4 | 77.0 | 84.8 |
| TDMI [14] | 90.0 | 91.1 | 87.1 | 81.4 | 85.2 | 84.5 | 78.5 | 85.7 |
| **JM-Pose (Ours)** | **90.7** | **91.6** | **87.8** | **82.1** | **85.9** | **85.3** | **79.2** | **86.4** |

**Table 2: Quantitative results on the PoseTrack2018 dataset.**

| Method | Head | Shoulder | Elbow | Wrist | Hip | Knee | Ankle | Mean |
|---|---|---|---|---|---|---|---|---|
| STAF [41] | - | - | - | 64.7 | - | - | 62.0 | 70.4 |
| AlphaPose [13] | 63.9 | 78.7 | 77.4 | 71.0 | 73.7 | 73.0 | 69.7 | 71.9 |
| TML++ [22] | - | - | - | - | - | - | - | 74.6 |
| MDPN [18] | 75.4 | 81.2 | 79.0 | 74.1 | 72.4 | 73.0 | 69.9 | 75.0 |
| PGPT [3] | - | - | - | 72.3 | - | - | 72.2 | 76.8 |
| Dynamic-GNN [64] | 80.6 | 84.5 | 80.6 | 74.4 | 75.0 | 76.7 | 71.8 | 77.9 |
| PoseWarper [5] | 79.9 | 86.3 | 82.4 | 77.5 | 79.8 | 78.8 | 73.2 | 79.7 |
| PT-CPN++ [68] | 82.4 | 88.8 | 86.2 | 79.4 | 72.0 | 80.6 | 76.2 | 80.9 |
| DCPose [34] | 84.0 | 86.6 | 82.7 | 78.0 | 80.4 | 79.3 | 73.8 | 80.9 |
| DetTrack [55] | 84.9 | 87.4 | 84.8 | 79.2 | 77.6 | 79.7 | 75.3 | 81.5 |
| SLT-Pose [16] | 84.3 | 87.5 | 83.5 | 78.5 | 80.9 | 80.2 | 74.4 | 81.5 |
| FAMI-Pose [35] | 85.5 | 87.7 | 84.2 | 79.2 | 81.4 | 81.1 | 74.9 | 82.2 |
| HANet [25] | 86.1 | 88.5 | 84.1 | 78.7 | 79.0 | 80.3 | 77.4 | 82.3 |
| M-HANet [24] | 86.7 | 88.9 | 84.6 | 79.2 | 79.7 | 81.3 | 78.7 | 82.7 |
| KPM [15] | 85.1 | 88.9 | 86.4 | 80.7 | 80.9 | 81.5 | 77.0 | 83.1 |
| TDMI [14] | 86.2 | **88.7** | 85.4 | 80.6 | 82.4 | 82.1 | 77.5 | 83.5 |
| **JM-Pose (Ours)** | **86.6** | **88.7** | **86.0** | **81.6** | **83.3** | **83.2** | **78.2** | **84.1** |

(elbow, knee): with an mAP of 87.8 (↑ 0.7) for the elbow and an mAP of 85.3 (↑ 0.8) for the knee. The consistent improvement of our method underscores the significance of explicitly incorporating context-aware joint feature and global pixel-level motion flow.

**Results on the PoseTrack2018 dataset.** We further evaluate the proposed JM-Pose on the PoseTrack2018 dataset. Empirical comparisons on the validation set are presented in Table 2. JM-Pose outperforms all other methods, achieving an mAP of 84.1, with an mAP of 86.0, 81.6, 83.3, and 78.2 for the elbow, wrist, knee, and ankle joints.

**Table 3: Quantitative results on the Posetrack21 dataset.**

| Method | Head | Shoulder | Elbow | Wrist | Hip | Knee | Ankle | Mean |
|---|---|---|---|---|---|---|---|---|
| Tracktor++ w. poses [4] | - | - | - | - | - | - | - | 71.4 |
| CorrTrack [42] | - | - | - | - | - | - | - | 72.3 |
| CorrTrack w. ReID [42] | - | - | - | - | - | - | - | 72.7 |
| Tracktor++ w. corr [4] | - | - | - | - | - | - | - | 73.6 |
| DCPose [34] | 83.2 | 84.7 | 82.3 | 78.1 | 80.3 | 79.2 | 73.5 | 80.5 |
| FAMI-Pose [35] | 83.3 | 85.4 | 82.9 | 78.6 | 81.3 | 80.5 | 75.3 | 81.2 |
| TDMI [14] | **85.8** | 87.5 | 85.1 | 81.2 | 83.5 | 82.4 | 77.9 | 83.5 |
| **JM-Pose (Ours)** | **85.8** | **88.1** | **85.7** | **82.5** | **84.1** | **83.1** | **78.5** | **84.0** |

**Table 4: Performance comparisons on the Challenging-PoseTrack dataset.**

| Method | Head | Shoulder | Elbow | Wrist | Hip | Knee | Ankle | Mean |
|---|---|---|---|---|---|---|---|---|
| SimpleBaseline [59] | 64.2 | 57.3 | 44.4 | 36.8 | 45.1 | 31.6 | 25.8 | 45.0 |
| HRNet-W48 [48] | 61.1 | 56.2 | 48.0 | 39.5 | 46.4 | 34.2 | 32.0 | 46.4 |
| PoseWarper [5] | 65.2 | 58.7 | 49.6 | 40.9 | 47.8 | 36.8 | 30.4 | 48.0 |
| DCPose [34] | 67.5 | 60.4 | 49.7 | 40.0 | 50.1 | 37.3 | 30.0 | 48.7 |
| FAMI-Pose [35] | 69.3 | 62.2 | 50.0 | 43.7 | 49.4 | 40.2 | 38.0 | 51.6 |
| TDMI [14] | 71.8 | 63.4 | 53.7 | 46.0 | 53.5 | 44.3 | 39.6 | 54.4 |
| **JMPose (Ours)** | **71.9** | **65.7** | **56.4** | **47.4** | **56.0** | **45.6** | **42.0** | **56.1** |

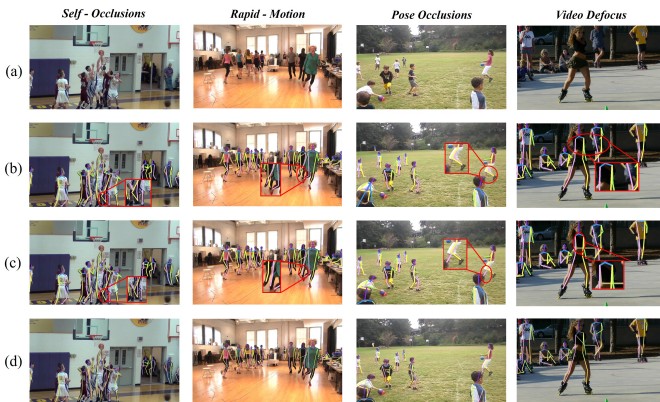

**Figure 4: The keyframe (a) and visual comparisons of detection results obtained from DCPose (b), TDMI (c), and our JM-Pose (d) on challenging scenes in the PoseTrack dataset. Inaccurate predictions are highlighted with the red solid circles.**

**Results on the PoseTrack21 dataset.** Table 3 shows that our JM-Pose achieves the best mAP performance on the PoseTrack21 dataset. Experimental results of the first four methods are officially provided by the dataset [10]. [14] offers the quantitive results of three methods (*i.e.,* DCPose [34], FAMI-Pose [35], TDMI [14]). We observe that the previous approach TDMI has delivered a commendable result. In comparison to TDMI, our JM-Pose improves the performance with an mAP of 84.0. Additionally, a notable observation from Table 3 is that JM-Pose achieves an mAP of 82.5 for the wrist joint and 78.5 for the ankle joint.

## 4.3 Performance on Complex and challenging scenes (RQ2)

We conduct experiments to evaluate the performance of our method and existing methods on complex and challenging scenes. We select a more difficult subset (Challenging-PoseTrack) accroding to [30] from the PoseTrack dataset [23]. Challenging-PoseTrack consists of 806 frames and contains many challenging scenes, such as pose occlusions, tangleed and crowded person, *etc.* Seven methods are evaluated in this subset and the quantitative results are presented in Table 4. The heatmap-based method DCPose tends to fail in these complex scenes, possibly due to the lack of the ability to capture spatio-temporal features, achieving an mAP of 48.7 only. The feature-based method TDMI integrates well-designed visual features by utilizing feature difference and obtains a mAP of 54.5. However, visual features fail to capture pixel-level motion dynamic and are not robust enough to challenging scenes. Our JM-Pose mutually amalgamates local joint dependency and global pixel-level motion flow to capture more informative representations, which effectively reason about challenging keypoints and improve model robustness. Therefore, our method achieves a final AP of 56.1 on the Challenging-PoseTrack.

## 4.4 Comparison of Visual Results (RQ3)

Looking at the visuals would be more instructive. We visualize the results for scenes with rapid motion or pose occlusions attest to the robustness of our method. Specifically, we present in Figure 4 the visual comparisons of our JM-Pose against state-of-the-art methods. From Figure 4, we can observe that JM-Pose displays more accurate pose estimation given the heavily crowded and challenging scenarios. DCPose only leverages temporal heatmap information from neighboring frames to refine the initial pose heatmap, resulting in suboptimal performance. TDMI solely refines and disentangles backbone features yet ignores the important effect of joint cues. For example, in the *Rapid-Motion* scene (c), the right foot and right knee (pink line) of the woman wearing black in the TDMI red box are to the left of the true keypoint location, while our method identified the correct location. Similarly, in the *pose occlusions* scene (c), the right knee (pink line) of the woman wearing white pants in the TDMI red box is located to the right of the true keypoint location. At the same time, TDMI did not estimate the right hip of the boy (in the lower right corner of the image), while our method identified the correct location. More visual results of our method are demonstrated in Figure 5 and supplementary material. Through the principled designs of context-aware joint learner and joint-motion mutual learning for mining complementary and informative representations, our JM-Pose is capable of detecting accurate human poses for challenging scenes.

## 4.5 Ablation Study (RQ4)

To better investigate the contribution of JM-Pose under different settings, we conduct a series of ablation studies, including context-aware joint learner (CJL) and joint-motion mutual learning (JMML). These experiments are conducted on the PoseTrack2017 validation dataset.

**Study on components of JM-Pose.** We conduct experiments to investigate the effect of each component in our JM-Pose and

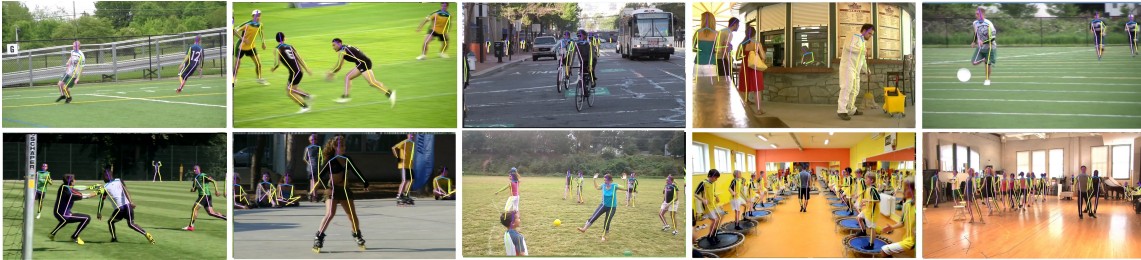

**Figure 5: More visual results of JM-Pose on benchmark datasets. Challenging cases such as high-speed motion, video defocus, and pose occlusion are involved.**

**Table 5: Ablation of different components of JM-Pose. 'w/o X' represents remove X module.**

| Method | CJL | JMML | Mean | Declines |
|---|---|---|---|---|
| JM-Pose | ✓ | ✓ | **86.4** | - |
| JM-Pose* | ✓ | ✓ | 85.9 | 0.5 (↓) |
| w/o CJL | - | ✓ | 85.6 | 0.8 (↓) |
| w/o JMML | ✓ | - | 84.6 | 1.8 (↓) |
| TDMI [14] | - | - | 85.7 | 0.7 (↓) |

**Table 6: Ablation of different modules in joint-motion mutual learning (JMML).**

| Method | $\mathcal{L}_{IO}$ | JMIB | Mean | Declines |
|---|---|---|---|---|
| JM-Pose | ✓ | ✓ | **86.4** | - |
| (a) | ✓ | - | 85.0 | 1.4 (↓) |
| (b) | - | ✓ | 85.8 | 0.6 (↓) |

present the empirical result in Table 5. TDMI serve as a baseline which refines visual features for human pose estimation. However, it neglects local joint dependency present in heatmaps. In addition, JM-Pose* refers to using visual feature differences to extract motion features (from backbone network), which replaces the global pixel-level motion flow in JM-Pose and the mAP decreases from 86.4 to 85.9. This performance drop (↓ 0.5) highlights that our global motion flow could capture pixel-level motion dynamics and is helpful for human pose estimation. In case w/o CJL, we remove the context-aware joint learner (CJL), and the initial heatmap $\hat{H}_t$ and motion flow $M_t$ are fed into the JMML. We can observe that mAP falls from 86.4 to 85.6. This decline in performance upon removal of the context-aware joint learner suggests the efficacy of the joint-level feature clues encoded by the context-aware joint learner. Moreover, in case w/o JMML, we remove joint-motion mutual learning (JMML), and the local joint feature $J_t$ is fed into the detection head. We can observe that mAP falls from 86.4 to 84.6 mAP. This decline in performance upon removal of the joint-motion mutual learning suggests the importance of the mutually aggregate the joint feature and motion flow.

**Study on components of JMML.** Finally, we also examine the contributions of joint-motion mutual learning (JMML) under different settings and report the results in Table 6. (a) For the first setting, we remove the joint-motion interaction block (JMIB) module, which performs mutual learning between joint feature and motion flow. It should be noted that after removing JMIB, our proposed information orthogonality objective ($\mathcal{L}_{IO}$) will constrain the joint feature and motion flow in the context-aware joint learner to generate diverse joint feature. The mAP results drop from 86.4 to 85.0. This performance deterioration on top of JMIB corroborates the importance of excavating the complementary joint-motion cues in guiding more accurate estimations of the joint locations. (b) For the next setting, we remove the information orthogonality objective ($\mathcal{L}_{IO}$) from JMML, which minimizes redundant information and differentiates the context-aware joint feature and global motion flow. We observe the performance drops from 86.4 mAP to 85.8 mAP. This demonstrates the importance of our $\mathcal{L}_{IO}$ objective in introducing meaningful and diverse joint feature and motion flow to enhance human pose estimation.

## 5 Conclusion

In this work, we explore human pose estimation from the perspective of effectively integrating context-aware local joint feature and global pixel-level motion flow. We propose a novel joint-motion mutual learning framework termed JM-Pose. We propose a context-aware joint learner guided by heatmaps to retrieve local joint-level information from the motion flow. A particular highlight of our method is the progressive joint-motion mutual learning to exchange information between joint feature and motion flow, improving the performance of pose estimation. Theoretically, we further propose an information orthogonality objective for effective feature diversity supervision of joint-motion interactive blocks. Empirical evaluation on three large-scale datasets shows that our method achieves notable improvements, showcasing the ability of our method to detect human poses in complex and challenging scenes.

## 6 Acknowledgments

This research is supported by the National Natural Science Foundation of China (No. 62276112, No. 62372402), Key Projects of Science and Technology Development Plan of Jilin Province (No. 20230201088GX), the Key R&D Program of Zhejiang Province (No. 2023C01217), and Graduate Innovation Fund of Jilin University (No. 2024CX089).

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
