# OpenReview forum: "Joint-Motion Mutual Learning for Pose Estimation in Video"
_acmmm.org/ACMMM/2024/Conference — MM2024 Poster_

### Official Review · Reviewer_8xGC · 2024-05-21

**Rating:** 3
**Confidence:** 3

**Summary:**

This paper proposed a joint-motion mutual learning framework for video pose estimation. Specifically, a context-aware joint learner is introduced to learn joint features between initial heatmaps and motion flow. In addition, a progressive joint-motion mutual learning structure is developed to fuse local join features and global motion flow. Experimental results show that the proposed method can outperform the state-of-the-art on several benchmark datasets.

**Strengths:**

1. The paper is well-written and easy to follow.
2. The proposed context-aware joint learner (CJL) and progressive joint-motion mutual learning (JMML) sound interesting.
3. The paper provides sufficient experimental comparisons with state-of-the-art methods on several benchmark datasets.

**Limitations:**

1. My major concern is the contribution of this paper. To me, the proposed model is more like a multi-modal feature fusion framework, and the core idea is how to fuse intra-frame spatial $\hat{H}_t$ and inter-frame temporal feature $M_t$, i.e., motion flow, for video pose estimation. So what is the key insight of this proposed model? What is the difference between the proposed model and existing works?

2. Could authors elaborate more on the definitions of “local joint dependency” and “global pixel-level motion dynamics”? How to define the concepts of “local” and “global” here? What is “global pixel-level motion dynamics”? I mean the motion flow is just calculated by two consecutive frames? Why is it global?

3. Figure 2(b) CJL block shows that the joint feature $\hat{H}_t$ and the motion flow $M_t$ are concatenated and then fed into two separate convolutional layers for producing the offset and convolution weights of the deformable convolution. I wonder why the motion flow should be modulated by using such a deformable convolution. What is the rationale behind this?

4. If the motion flow $M_t$ should be modulated, then the $J_t$ will be regarded as another form of motion cues. Therefore, the inputs of JMML are from the same modality, but it means no joint features are fed into the JMML. I wonder how to perform “joint-motion” mutual learning in this setting.

5. The notations in Figure 1 are confusing. For example, $H_{t-1}$ denotes the extracted feature of $I_{t-\delta}$, so as the $H_{t+1}$ and the $I_{t+\delta}$. As mentioned in the paper, here $\delta = 1$. But the figure shows that the $\delta$ has suddenly disappeared. In addition, $M_t^r$ and $M_t^l$ are not well-explained and not mentioned in the main text, maybe the superscript $r$ and $l$ refer to “right” and “left” motion flow, respectively?

6. For the JMML, Sec 3.2, could authors elaborate more on why the information orthogonality objective should be achieved in the multi-modal feature fusion? The motivation for introducing such information orthogonality objective is not clear to me.

**Suitability:**

2

---

### Official Review · Reviewer_nHkG · 2024-05-24

**Rating:** 5
**Confidence:** 2

**Summary:**

This paper proposes a novel joint-motion mutual learning framework for human pose estimation in videos, which addresses the limitations of existing methods that fail to leverage both local joint information and global motion dynamics. The framework introduces a context-aware joint learner that utilizes initial heatmaps and motion flow to extract robust joint features. A progressive joint-motion mutual learning mechanism is further proposed to enhance the model's performance by synergistically exchanging information between joint features and motion flow. An information orthogonality objective is introduced to prevent learning redundant information. Experimental results show the proposed method outperforms previous works on challenging benchmarks.

**Strengths:**

This article proposes a new method for joint training and attitude estimation using motion features in videos. Although the idea of using motion features is not novel, the author fully considers the modeling relationship between motion and joints, and uses deformable conv to achieve effective feature fusion between joints and motion, thus obtaining good experimental results.

**Limitations:**

In my opinion, there are no significant flaws or shortcomings in this paper. However, due to the reliance of the paper on motion+heatmap, a classic solution in video pose estimation, its novelty is undoubtedly limited. The hypothesis and framework of motion+joint advocated by the author may seem reasonable, but there are still some rough details, such as the symmetrical model design of JMML, which integrates motion and joint features as the same type of features, abandoning the relationship modeling between joints and motion constructed using DCN earlier. I believe that JMML should have a better architecture design.

**Suitability:**

3

---

### Official Review · Reviewer_L1wn · 2024-05-25

**Rating:** 3
**Confidence:** 2

**Summary:**

This paper proposes a joint-motion mutual learning framework for pose estimation, which effectively concentrates on both local joint dependency and global pixel-level motion dynamics. It introduces a context-aware joint learner that adaptively leverages initial heatmaps and motion flow to retrieve robust local joint feature. Empirical experiments show this method outperforms prior arts on three challenging benchmarks.

**Strengths:**

- The paper proposes a novel method by integrating both heatmap-based and feature-based paradigms to address the challenges (e.g. video defocus and self-occlusion) in complex video scenes for video pose estimation.
- Extensive analysis and visualization of existing paradigms in Fig.1.
- Anonymous codes are provided.

**Limitations:**

- Its contributions, e.g. CJL and JMML, are limited to several slight technical modifications and combinations to the well-established technology in video domain (Motion-Appearance Interaction via transformer, gated weight and deformable operations).
- It seems that no component is specifically designed for pose prediction or joint. It lacks significant distinction from appearance-motion mutual learning frameworks designed for other video tasks.

Minor Limitation
- The local joint dependency means the temporally local information among consecutive frames or spatially local information within each frame, stating this information in the Introduction for better readability. What are the differences between the proposed method and the common Image (spatial)/Optical Flow (temporal) based methods?
- Adding several multi-modality fusion methods (Appearance-Motion) of video tasks in Related work can help to understand the differences from other methods.
- There should be more detailed labels in Fig.2, such as JO and JW. The related information should be available in the figures without reading corresponded contents.
- RAFT can generate high-quality flow map, while you can use PWCNet or FlowNet to verify the model's robustness and generalization ability with low-quality optical flow.

**Suitability:**

1

---

### Official Review · Reviewer_eMdg · 2024-06-01

**Rating:** 4
**Confidence:** 3

**Summary:**

This paper aims to improve human pose estimation in videos, based on the hypothesis that local information (about human joint positions) and global information (about motion dynamics) are complementary, and both are crucial to accurate estimation results, particularly in challenging scenarios.
A joint-motion mutual learning framework is proposed to combine useful information in joint detection heatmaps ("local") with dense optical flows ("global"), which utilizes modulated deformable convolutions and stacked cross-attention blocks.
Experimental results demonstrate its superiority over previous methods that only leverage either "local" or "global" features.

**Strengths:**

**Novelty and clarity**:
The observations and the ideas of leveraging the complementarity of local joint feature and global motion flow are somewhat novel.
The comparison between heatmap-based methods and feature-based methods in Figure 1 clearly presents the motivation and the philosophy.

**Significant results**:
Quantitative results presented in Table 1-4 show that the proposed method significantly outperforms previous state-of-the-art methods in various scenarios by a large margin. The visual comparisons also demonstrate the weaknesses of previous methods and the strengths of the proposed one.

**Limitations:**

**Insufficient evidence**:
Though the evaluation results show that the proposed method outperforms previous studies notably, it has not yet been proved that these components (more specifically, JMIB blocks) work as expected, i.e., "progressively exchanging valuable information between joint feature and motion flow, hence enhancing both branches orthogonally and comprehensively".

More evidence is required to show that these $M_t^i$ and $J_t^i$ are indeed progressively improved, and a stack of $L$ consecutive blocks is essential for this progressive process, rather than just enlarging model capacity.

For example, the performances of training detection heads on intermediate blocks $l=1,...,L-1$ could be reported to show a progressive refinement process.
Additionally, is the $L_{IO}^i$ value (or other metrics) progressively decreasing when $i$ increases from $1$ to $L$?
Visualizations of $M_t^i$ and $J_t^i$ may also be helpful if they are meaningful.

**Comparison fairness**:
Moreover, the information about model capacity (e.g., parameter count, inference speed, flops) is missing when comparing to other methods. Since the proposed method involves new operations like deformable convolution and cross-attention, such information is required to ensure a fair comparison.

**Suitability:**

2

---

### Meta-Review · Area_Chair_LQTQ · 2024-07-03

**Recommendation:** Accept (Poster)
**Confidence:** 5

**Metareview:**

Reviewers pointed out that the observations and the ideas of leveraging the complementarity of local joint feature and global motion flow are novel. The designed context-aware joint learner (CJL) and progressive joint-motion mutual learning (JMML) scheme are interesting.
This work also provides extensive experimental comparisons with state-of-the-arts on benchmarking datasets. All the reviewers thus recommend acceptance. AC agrees with the reviewers.